# COLLECTIVE VARIABLES OF NEURAL NETWORKS: EMPIRICAL TIME EVOLUTION AND SCALING LAWS

## ABSTRACT

This work presents a novel means for understanding learning dynamics and scaling relations in neural networks. We show that certain measures on the spectrum of the empirical neural tangent kernel, specifically entropy and trace, yield insight into the representations learned by a neural network and how these can be improved through architecture scaling. These results are demonstrated first on test cases before being shown on more complex networks, including transformers, auto-encoders, graph neural networks, and reinforcement learning studies. In testing on a wide range of architectures, we highlight the universal nature of training dynamics and further discuss how it can be used to understand the mechanisms behind learning in neural networks. We identify two such dominant mechanisms present throughout machine learning training. The first, information compression, is seen through a reduction in the entropy of the NTK spectrum during training, and occurs predominantly in small neural networks. The second, coined structure formation, is seen through an increasing entropy and thus, the creation of structure in the neural network representations beyond the prior established by the network at initialization. Due to the ubiquity of the latter in deep neural network architectures and its flexibility in the creation of feature-rich representations, we argue that this form of evolution of the network's entropy be considered the onset of a deep learning regime.

## 1 INTRODUCTION

Scaling behaviour in neural networks has become a pivotal area of investigation in modern deep-learning research. Traditional scaling laws, which correlate the number of network parameters with performance metrics, provide a foundational understanding but often fall short when comparing different architectures or capturing the intricacies of neural network dynamics arising from the architecture of the network itself. To bridge this gap, our research leverages the neural tangent kernel (NTK) to elucidate neural networks' learning dynamics and scaling laws, offering a more comprehensive framework for understanding how network representations evolve and how architecture scaling impacts these processes. Specifically, this investigation utilizes a set of collective variables, the entropy and trace of the empirical NTK, to identify and explain learning regimes during the training process. The NTK, introduced by Jacot et al. (2018), provides a powerful tool for analyzing the infinite-width limits of neural networks. By focusing on the empirical NTK and its evolution during training, we can gain insights into the internal representations learned by finite-size neural networks and their dependence on architecture. Our approach is validated across various tasks and network types, including transformers, auto-encoders, graph neural networks, and reinforcement learning models, demonstrating its broad applicability and robustness.

### 1.1 STATE OF THE ART

Previous work has established various scaling laws for neural networks, often relating to the number of parameters and the expected performance (Bahri et al., 2024). However, these descriptions are predominantly phenomenological and do not fully capture the dynamics of different architectures or the nuances of the learning process. In Bahri et al. (2024), neural scaling laws are studied in a so-called resolution-limited and variance-limited regime, highlighting changes in the network performance within each. While this work highlights a useful relationship between the data manifold

and neural scaling, it does not address the fundamental question of whether learning mechanisms within each regime change at all. Leclerc & Madry (2020) identify regimes during training and relate them to a learning-rate dependent mechanism, namely, large-step and small-step regimes. They highlight that the process by which the network learns information differs between these two regimes and can be leveraged more effectively when considered as two separate training phases. In their 2021 paper, Geiger et al. (2021) explored the role of initialization in over-parameterized neural networks and how it changes the training process of the neural network. The problem has also been explored with more practical uses in finding, for example, in Mirzadeh et al. (2020) learning regimes, as impacted by batch size, learning rate, and regularization, is explored in relation to catastrophic forgetting, or the seminal work of Kaplan et al. (2020) where scaling laws are introduced as a means to optimize a cost-performance tradeoff in large language models.

The NTK has emerged as a key concept for understanding these dynamics, particularly in the context of wide neural networks where it facilitates a tractable analysis of gradient descent dynamics. Often, this work studies alignment effects in the NTK and what role these play in the learning taking place (Atanasov et al., 2021), or in identifying dynamics regimes (Lewkowycz et al., 2020). Indeed, a landmark result utilizing theoretical scaling laws studies came in the form of $\mu$-transfer (Yang & Hu, 2022; Yang et al., 2022), that can be used to initialize width-scaled networks without losing accuracy.

Each of these studies highlights the importance of scaling laws and network dynamics in the field's current state. However, in each case, they do so by studying raw loss metrics as a function of changing network size and identifying steady changes in their performance, which are characterized by a scaling law. In this work, we explore an alternative means of representing training dynamics, namely, the collective variables of the NTK. In doing so, we isolate fundamental learning mechanisms such as information compression and structure formation and highlight their role and evolution during training.

## 1.2 CONTRIBUTION STATEMENT

We identify several relevant contributions in this work:

- We provide a detailed empirical analysis of NTK evolution across various neural network architectures and tasks, highlighting universal patterns and architecture-specific behaviors.
- We introduce collective variables derived from the NTK, such as entropy and trace, to quantify the diversity and effective learning rates of neural network updates.
- We demonstrate the applicability of our methods to modern machine learning models, including transformers, graph neural networks, and reinforcement learning agents, showcasing the utility of NTK-based analysis for understanding complex learning dynamics.
- We find that large models universally show an increase in entropy, in contrast to small models. This allows us to quantitatively define a deep learning regime.
- We relate existing scaling parameters of width and depth with our entropy and trace collective variables.

By extending the analysis of NTK to encompass a wide range of network architectures and tasks, we offer new tools and perspectives for scaling and comparing neural networks at an information-theoretic level. This work enhances our theoretical understanding of neural network dynamics and provides practical insights for optimizing architecture design and training protocols.

## 2 THEORY AND METHODS

### 2.1 NEURAL TANGENT KERNEL FOR NETWORK DYNAMICS

The neural tangent kernel was first introduced to the machine-learning community by Jacot et al. (2018) to investigate the infinite-width limits of neural networks. In this work, we explore the evolution of neural networks and how this can be better understood by studying measures on this matrix. However, to understand why this is possible, consider a neural network, $f[x_i, \boldsymbol{\theta}]$ acting on

a single input data point, $x_i$ and parametrized by $\boldsymbol{\theta}$. During training, each element of parameter, $\theta_k$, will be updated via some variant of stochastic gradient descent

$$\theta_{k,t+1} = \theta_{k,t} - \eta \sum_i \frac{\partial \mathcal{L}(f[x_i, \boldsymbol{\theta}_t], y_i)}{\partial \theta_k}, \tag{1}$$

where $\eta$ is the learning rate, $\mathcal{L}$ is the chosen loss function used in the minimization, and $y_i$ is the target value associated with the input data, $x_i$. When performing gradient descent, the size of the step forward is determined by the learning rate, which is set under the assumption that the loss surface will not change drastically over the course of the update. However, to perform a more rigorous analysis, we take this further and move into continuous time by defining

$$\dot{\theta}_k = \lim_{\eta \to 0} \frac{\theta_{k,t+1} - \theta_{k,t}}{\eta} = -\sum_i \frac{\partial \mathcal{L}(f[x_i, \boldsymbol{\theta}_t], y_i)}{\partial \theta_k}, \tag{2}$$

where the dot denotes a time derivative. The problem with this formalism is that the network architecture itself is not expressed in the evolution, only the parameters of this architecture. To understand how the neural network representations evolve, we compute the time derivative of the function itself, $\dot{f}[x_i, \boldsymbol{\theta}]$ by the chain rule

$$\dot{f}[x_p, \boldsymbol{\theta}] = \sum_k \frac{\partial f[x_p, \boldsymbol{\theta}]}{\partial \theta_k} \dot{\theta}_k. \tag{3}$$

Substituting Equation 2 into Equation 3 and expanding the loss derivative into a parameter and function component, we find

$$\dot{f}[x_p, \boldsymbol{\theta}] = -\sum_k \frac{\partial f[x_p, \boldsymbol{\theta}]}{\partial \theta_k} \sum_i \frac{\partial f[x_i, \boldsymbol{\theta}_t]}{\partial \theta_k} \cdot \frac{\partial \mathcal{L}(f[x_i, \boldsymbol{\theta}_t], y_i)}{\partial f[x_i, \boldsymbol{\theta}_t]}. \tag{4}$$

Letting $\frac{\partial f[x_p, \boldsymbol{\theta}_t]}{\partial \theta_k} \cdot \frac{\partial f[x_i, \boldsymbol{\theta}]}{\partial \theta_k} = \Theta_{pi}$, and $\frac{\partial \mathcal{L}(f[x_i, \boldsymbol{\theta}_t], y_i)}{\partial f[x_i, \boldsymbol{\theta}_t]} = \partial_{f_i} \mathcal{L}_i$, we find

$$\dot{f}[x_p, \boldsymbol{\theta}] = \sum_i \Theta_{pi} \partial_{f_i} \mathcal{L}_i, \tag{5}$$

where the matrix $\Theta_{pi}$ is referred to as the neural tangent kernel or NTK. In the evolution of the neural network representations, the loss derivative term plays the role of the minimization constraint, guiding the network parameters into an optimum. The NTK, however, also plays a role in how the network evolves during training, specifically, how the architecture impacts this evolution. Unfortunately, the NTK is often very large, and therefore, alternative tools are required to study it and its evolution.

## 2.2 Collective Variables

The NTK matrix is built from the inner products of the gradient vectors of a neural network evaluated at different data points. It provides information on the degree of correlation in these gradient vectors, i.e., whether the representations of these points will evolve similarly or not. To see the role this plays in the evolution, consider the eigendecomposition of the NTK

$$\Theta_{pi} = \sum_n D_{pn} \Lambda_{nn} \left( D^{-1} \right)_{ni}, \tag{6}$$

where $D_{pn}$ is the matrix of eigenvectors and $\Lambda_{ij}$ is the diagonal matrix of eigenvalues. Substituting Equation 6 into Equation 5, yields (see also Krippendorf & Spannowsky (2022))

$$\dot{f}[x_p, \boldsymbol{\theta}] = \sum_i \sum_n D_{pn} \Lambda_{nn} \left( D^{-1} \right)_{ni} \partial_{f_i} \mathcal{L}. \tag{7}$$

Evaluating Equation 7 from right to left, we can see that during an update of the network, the loss derivative of the $i^{\text{th}}$ data-point is projected along the eigenvectors of the NTK and scaled by its eigenvalues, thus introducing effective learning rates and direction changes due solely to the architecture and its instantaneous prior over the data. This description can be simplified into

$$\dot{f}[x_p, \boldsymbol{\theta}] = \sum_n D_{pn} \tilde{p}_n, \tag{8}$$

| Name | Task | Architecture | Citation |
| --- | --- | --- | --- |
| UD Treebank Ancient Greek | NLP | Transformer | (Eckhoff et al., 2018; Bamman & Crane, 2011) |
| OGBG MolPCBA | Classification | GNN | (Hu et al., 2021) |
| Atari | RL | CNN | (Bellemare et al., 2013), |
| CIFAR10 | Classification | ResNet18 | (Krizhevsky & Hinton, 2009) |
| Fuel Efficiency | Regression | FFNN | (Quinlan, 1993) |
| MNIST | Generative | FFNN / CNN | (Lecun et al., 1998) |

Table 1: Data-sets along with the model architecture, task, and citation of the different experiments used throughout the study.

where $\tilde{p}_n = \lambda_n D_{ni}^{-1} \partial_{f_i} \mathcal{L}$, referred to here as the $\lambda$-scaled projection factor, acts as a scaling factor on the update to the representation of data point $x_p$. Regarding learning, this formalism argues that a neural network starts with a prior of the data, indicated by the spectrum of the NTK matrix, which correlates and scales modes based on its architecture. During an update, the fundamental modes present in the neural tangent kernel are tuned based on the loss function. This can occur either via scaling from the magnitude of the loss gradients, strengthening or weakening the degree of correlation between modes. Alternatively, reorientation via the dot product of the gradient vector can force a change in the direction of the eigenvectors. Thus, two data points that the network considered correlated before training, highlighted by high similarity in the NTK matrix, can either be separated via an orthogonal gradient vector or strengthened via an aligned gradient with a large eigenvalue. The question remains: how can we discuss this mechanism taking place inside the neural network? In their 2023 paper, Tovey et al. (2023) introduced measures on the NTK to describe its current state and used this description to describe the role of data selection in neural network training. By measuring the collective variables of their networks before training, they showed that it could provide insight into the quality of the prior data distribution and, thus, the expected performance of the training. Specifically, they introduced the entropy of the NTK, computed from the normalized eigenvalues as

$$S = -\sum_i \lambda_i \log \lambda_i \tag{9}$$

providing a measure of the diversity of the matrix, i.e., how many independent degrees of freedom exist in the update, and the trace of the NTK,

$$\sum_i \Theta_{ii}, \tag{10}$$

describing the magnitude of the scaling factor on the learning rate. This work extends their analysis to the evolution of the neural network during training to understand what processes it undergoes.

### 2.3 DATASETS AND ARCHITECTURES

Several datasets have been studied to demonstrate the use of the collective variables for model interpretation and to better highlight the dynamics' universality. Table 1 outlines each dataset, including their citations, the type of problem being learned, and the network architecture used in the training. Training for each network required unique optimizers, learning rates, and batch sizes and was performed over different data sets. The training was distributed on several compute clusters utilizing NVIDIA 4090, 3090, and L4 GPUs.

### 2.4 ZNNL

Most network training and evaluation are performed using the ZnNL Python library throughout this work. ZnNL utilizes the neural-tangents library (Bradbury et al., 2018) for NTK calculations and the Flax library (Heek et al., 2023) for neural network definition and training. The novelty methods were trained using the Flax library using customized scripts outside the ZnNL framework.

# 3 Collective Variable Evolution

This work aims to understand the learning dynamics of neural networks and identify any universal properties of this evolution. To this end, networks of different architectures are trained on different datasets and their properties, entropy, trace and loss, are measured at each epoch. This has been performed on standard classification and regression problems in simple networks but also scaled to more modern architectures and data structures, including generative models, language problems with transformers, graph neural networks, and reinforcement learning. First, we discuss the most comprehensive results of the MNIST data scan; afterwards, we introduce the additional architectures in more detail.

## 3.1 Architecture Scans

In the first study, the MNIST dataset is used to train dense and convolutional neural networks, along with the Fuel Efficieny dataset used to train a dense network, the architectures of which are outlined in Table 1. Figure 1 displays the evolution of the collective variables for certain combinations of widths, depths, and activation functions for the convolution model. Further results for dense MNIST and the Fuel Efficiency task are shown in Appendix figures 4 and 5 respectively, but are qualitatively similar. Each plot shows the variables computed on the test and the train data, differentiated by colors. Furthermore, each row corresponds to the network's varying architecture property while the other properties are held fixed, similar to performing measurements in a specific ensemble in statistical physics. In addition to the evolution diagram, a more comprehensive scaling study was performed on the dense network architecture. This was done by training dense neural networks of varying depths and widths and computing the loss, entropy, and trace evolution. The results of this study at the start of training and after 100 epochs are displayed in Figure 2. To fully digest these results, we will discuss each variable individually, beginning with the loss.

**Loss:** In the case of both test and training loss, we see drastic improvements with changes to the architecture. Namely, using deeper networks, TanH over ReLU activation functions, and training wider networks results in lower test and train losses. This trend is broken in the depth scaling as we see the test loss, and therefore, generalization improves with increasing depth despite a larger train loss, in line with previously reported scaling behaviour (Yang & Hu, 2022; Bahri et al., 2024). These results are further strengthened in Figure 2 b) in the case of a dense network, as we see the deeper networks achieving significantly improved losses compared with their shallower counterparts.

**Entropy:** Entropy evolution highlights two distinct learning dynamics that relate to different learning regimes arising directly from the information content in the neural network. In general, entropy undergoes one of two evolutions: information compression and structure formation. Information compression is signified by a reduction in the NTK's entropy, which indicates that gradient vectors are beginning to align and the network is relating many data points to one another. An increasing entropy indicates structure formation, suggesting that the network separates data to identify better representations. What we observe in Figure 1 and in the dense network counterpart in Figure 4 is that depending on the size of the network, a different mechanism will become dominant. In most cases, we see the entropy drop at the beginning of training, coinciding with the sharp drop in the loss early in the optimization procedure. For smaller networks that are limited in dimension and, therefore, have limited flexibility to construct complex representations, the entropy remains low throughout the rest of the training. These networks also typically attain a worse train and test loss than their larger counterparts, although they can still train. After the compression phase of training, the larger networks undergo structure formation, indicated by increasing entropy. This relationship, however, becomes complex when depths vs width are considered. It is clear from the results that increasing the network dimension via width increases the entropy of the models. However, adding layers appears to reduce this entropy. Intuitively, this makes sense as adding layers does not increase the projected dimension but applies additional non-linearities to the representations. However, we also see that deeper networks can achieve higher entropies than their shallower counterparts when scaled further in width and after training. Thus, while adding layers to a network of one width can reduce its entropy, scaling it to a higher width increases its capacity for a higher entropy. To explore this mechanism further, the higher resolution architecture scans of the dense neural network architecture are presented in Figure 2 a) at the start of training and Figure 2 b) at the end. In these plots, increas-

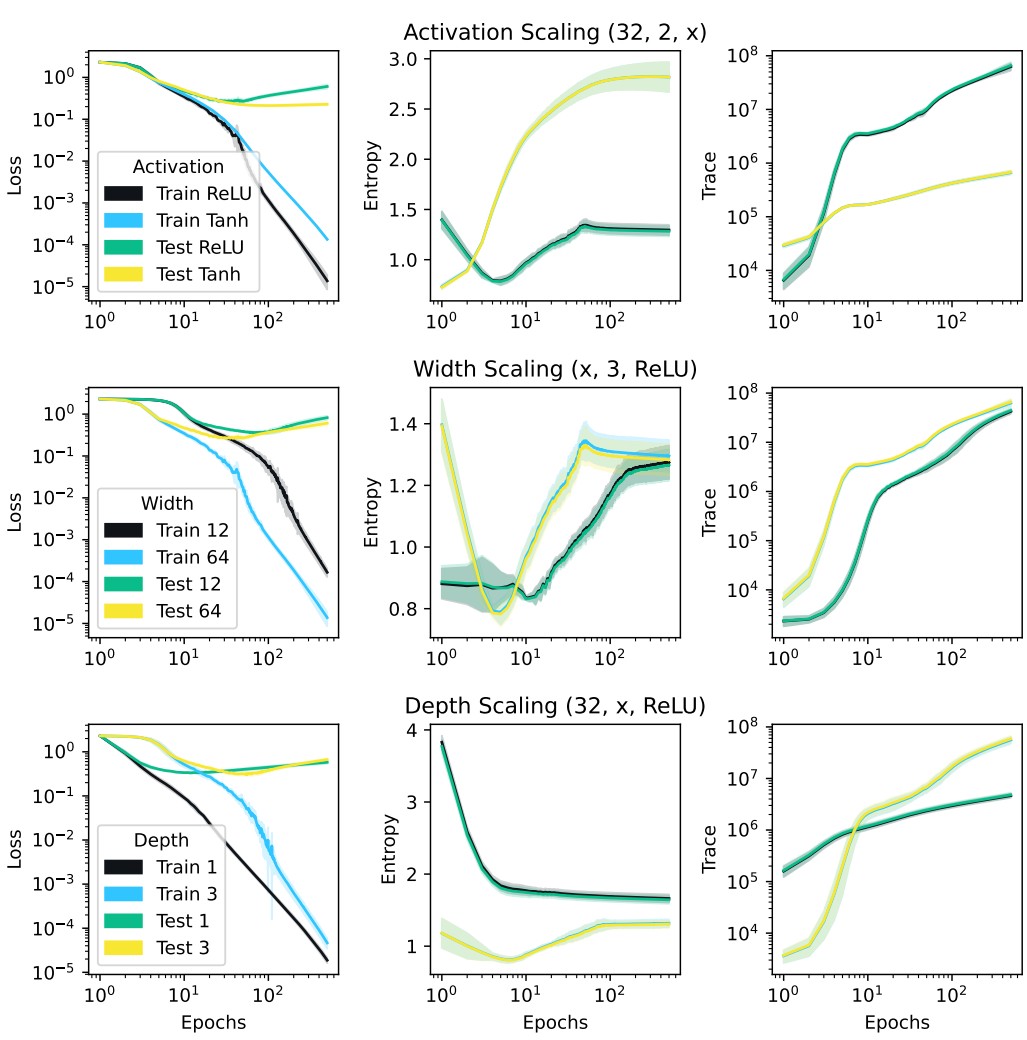

Figure 1: Time evolution diagrams for the convolutional MNIST study. The top row shows the evolution difference due to a changing activation function. The second row is width scaling, and the third is depth scaling. The tuple over the plots highlights the architecture being studied. It is structured as (Width, Depth, Activation), where an $x$ indicates that this property is being changed during the study.

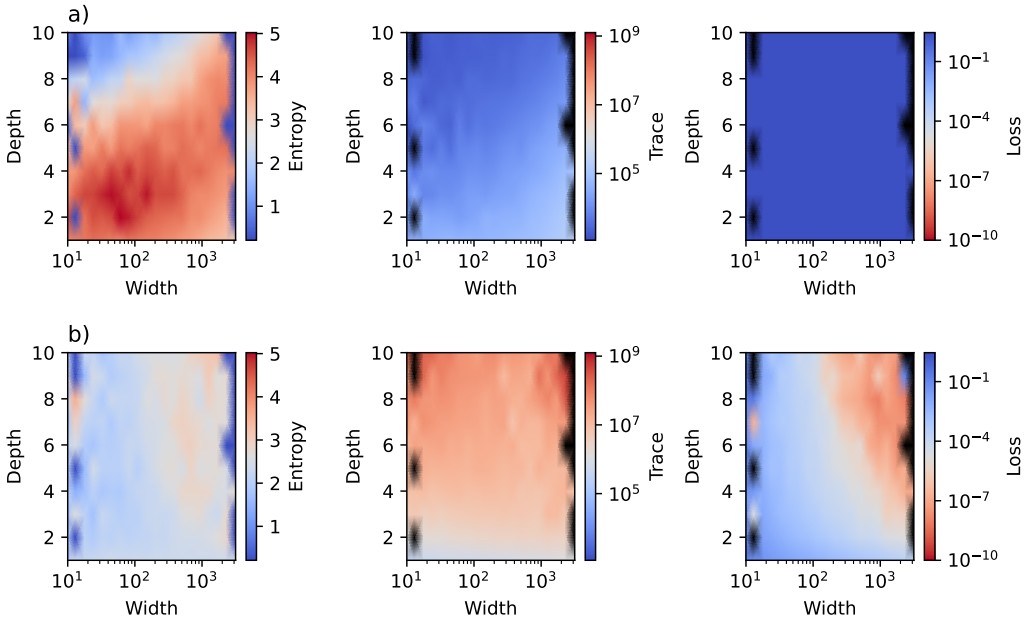

Figure 2: Architecture scaling sweep of a dense neural network trained on the MNIST dataset. The top row a) shows the entropy, trace and loss as a function of network width and depth at initialization. The bottom row, b), shows the same data after training the model.

ing the network depth reduces entropy at initialization. However, the deeper networks form a region of high entropy after training, corresponding with a better test loss. This trend is not reproduced in the TanH case, indicating that these mechanisms are sensitive to the activation function. Further work could explore how the scaling behavior for each activation can be used to perform targeted architecture selection.

**Trace:** The final property to study is the trace of the networks. Recall the trace corresponds to an effective learning rate, e.g., how far we will be pushed along a specific direction due to the network architecture. In each study, trace increases with training time to a very large value, often in step with the loss evolution. From a learning perspective, this indicates that the network continually increases its effective steps along specific directions as training goes on, similar in concept to the network's confidence that the solution it is forming is correct. From a physical perspective, this is very similar to the evolution of kinetic energy asymmetrically to the decrease in a potential. Indeed, if one were to consider the kinetic energy of a network representation $f(x_i)$, as $E^k = \frac{1}{2m_i}|\dot{f}(x_i)|^2$, a simple expansion of variables would show

$$E^k = \frac{1}{2m_i}\Theta_{ii}\sum_p \left(\frac{\partial \mathcal{L}(x_p)}{\partial \theta_j}\right)^2,\tag{11}$$

where $m_i$ is some point mass and $\Theta_{ii}$ is indeed the trace of the NTK plotted here. Representation of this value as kinetic energy is a convenient metaphor for a real effect within a neural network during training. Namely, as one minimizes the loss, the network appears to become very sensitive to changes in these losses, as evidenced by this effective scaling parameter becoming large. This indicates that if fictitious data were added to the data set late in training, driving the loss term up, the network would train on this data point with a significant degree of confidence as opposed to at the start of training, where it could possibly be ignored or saturated out by the other data points. The conclusion here is that adding malicious data or even very new data to a data set at the end of training will result in a larger change in parameters than it would at the start of training, even if the loss computed on this value were the same at both times, simply due to the state of the neural

network. This result has significant ramifications for adversarial attacks on trained models and may help explain why small additions of noise to a model can lead to a drastic change in its outputs. Further investigation of this property may lead to a reduction in model sensitivity and could also be applied to more effective continual learning strategies. In the architecture sweeps shown in Figure 2, we see a similar trend in the trace at initialization and, after training, only shifted. Specifically, increasing depth and width results in a larger trace value further exasperated by the training process. Thus, supporting the work of Yang & Schoenholz (2017), deeper and larger networks are more sensitive than their shallower counterparts, not just at initialization but also after training.

## 4 Novelty Models

While exploring the evolution of collective variables on test problems provides deep insight into scaling relations and dynamics, it does not directly relate to the current state of machine learning. To extend our analysis and argue that these dynamics are, in fact, a universal phenomenon, we apply this analysis to several so-called novelty models taken from various fields of machine learning. The results of these evolution studies are outlined in Figure 3 and discussed below. It should be noted throughout this discussion that due to the size of these architectures and data sets, the collective variables have not been computed at each epoch but rather at steps of between 10 and 1000 epochs. Further, the NTK matrix was often subsampled during the calculations using 200 samples of 20 data points to construct a measurement. Before discussing the individual models and the unique features emerging in the dynamics, their measurements' global properties appear consistent with the test cases presented above. In all cases, the entropy of the models increases throughout training, indicating that we are always within a structure formation regime. This is not unusual, as each network is far larger than those required to induce constrained, compressive learning. A notable difference is entropy's complete lack of a compression phase. This is likely because the dataset size is such that after even a single batch, so much back-propagation has been performed that the network moves directly to a structure formation phase. Such a process indicates that the parameters of the network are free to perform sophisticated structure formation, thus learning fundamental features of the data. Given the dominance of the entropy increase in the latter portion of the training, it is an open question whether the initial decrease is a required process or simply an artifact of non-ideal initialization. One explanation would be that those models that perform only compression are not performing what the community would call deep learning. Only those models that can perform this structure formation should be considered to be performing deep learning. Further, the trace in the later time consistently increases, in line with previous measurements. Below, we examine four models more closely and interpret some emergent features.

### 4.1 NLP

In the NLP problem, a transformer was trained to perform part-of-speech tagging for ancient Greek text from Eckhoff et al. (2018); Bamman & Crane (2011). We see a story similar to the test cases discussed in the entropy and trace evolution. The trace increases steadily as the loss decreases. The difference in magnitudes is interesting compared with the previous examples. While the accuracy of the transformer reached 99 %, the loss, due to the specific one chosen, does not get too small, resulting in a more reasonable trace value and, thus, less "confidence" during the updates.

### 4.2 Reinforcement Learning

The next novelty model study looked at how the entropy and the trace of an agent trained via deep actor-critic reinforcement learning evolved. To compute the NTK, a dataset of environments was constructed by letting the agent play the game hundreds of times at three stages of learning: un-trained, occasionally successful, and end of training. In this way, the dataset on which the NTK is computed covers the full range of possible configurations the agent could see. The entropy, trace, and reward curves in Figure 3 are all plotted using a log-x scale to highlight exactly where certain transitions in this evolution occur. The reason for this is the sharp increase in the entropy and trace after approximately 1000 training episodes. What is interesting about this transition is that the entropy and trace, particularly of the actor, increase in the episode before the reward of the agent follows suit. Upon finer examination, we see that this jump occurs in the episode directly before the agent begins to be able to perform the task successfully and achieve positive rewards. This indicates

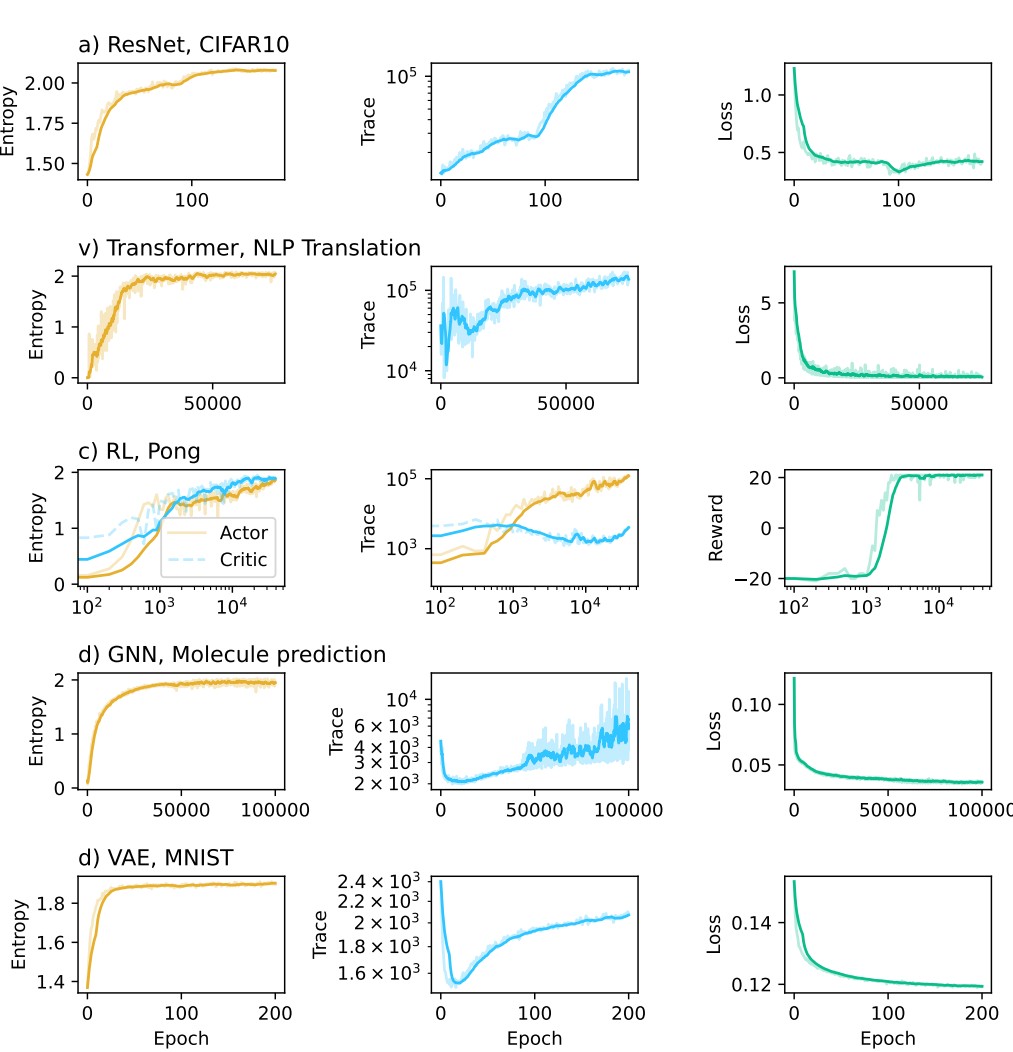

Figure 3: Collective variable evolution for the novelty architectures. In each case, the raw value, along with a running average, is shown. (a) A resnet18 trained on the CIFAR10 data set. (b) A transformer trained on part-of-speech tagging of ancient Greek. (c) A CNN-based reinforcement learner trained on the arcade game Pong. Note the use of reward instead of loss on the y-axis. (d) A graph neural network trained on molecular property prediction. (e) A simple variational auto-encoder trained to reproduce the MNIST dataset.

that the entropy, and to a lesser extent, the trace, indicated that the agent could perform its task better given its current set of parameters. This is a very natural interpretation of structure learning in a neural network, as seen through the eyes of an agent playing a game. At some stage during training, the neural network begins to identify structure in the data and can use this structure to make better moves in the game. The training then continues very quickly, partly due to the onset of structure formation but perhaps aided by the new scaling from the trace, which increases more significantly in the actor model. In the case of RL, the concept of confidence also becomes quite interpretable as the agent begins to trust more that the structure it is beginning to learn will be successful. Therefore, it can make larger steps in its updates, arising from the larger trace.

### 4.3 Graph Prediction and Generative Modelling

In graph prediction and generative models, similar dynamics are realized during training. Interestingly, in both cases, due to the small loss range, the trace evolution can be observed with a greater resolution, resulting in a large dip at the beginning of training. This dip occurs over a small range compared with the fluctuations and values observed in other models, but it is by no means a simple fluctuation. However, under the kinetic energy framework introduced above, where these dips arise could be accounted for by a missing term in the energy equation, perhaps related to repulsion between data points.

## 5 Conclusion

We have motivated the use of entropy and trace of the empirical neural tangent kernel as suitable measures of a neural network's state. These variables were then applied to understand architecture scaling and evolution on test problems in image classification and regression using dense and convolutional networks of various widths and depths. We identify two dominant regimes in entropy during learning: compression, where entropy decreases and latent space representations begin to collapse, and a structure formation regime, where the entropy increases again later in training, and the network better differentiates the data. We further identified that the compression regimes sometimes dominated the learning process in smaller architectures, albeit with degraded performance compared to the structure-forming networks. This quantitative handle to distinguish between the small and deep networks provides us with a classification of what it means to be in the deep learning regime. Further, we highlighted the trend of increasing trace during training, particularly in late time evolution. As shown, this trace can be associated with an effective learning rate during model updates, thus indicating that adding data with a large loss to the network late in training could result in a very sharp change in parameters despite the model being otherwise well-trained. To demonstrate the further use of these variables, we study the collective variable evolution on a set of so-called novelty models, which are more closely aligned with the current state of the art. In each case, we see a similar trend in the entropy and trace, with a notable, short-time difference in the trace of the autoencoder and graph neural network studies. Overall, the dynamics observed appear universal among architectures and shed light on how neural networks collect data in their latent space. Further, we indicate avenues for improved stability by exploring the reduction of trace late in training.

## 6 Ethics Statement

The authors declare no conflicts of interest or other ethical concerns.

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

# 7 APPENDIX

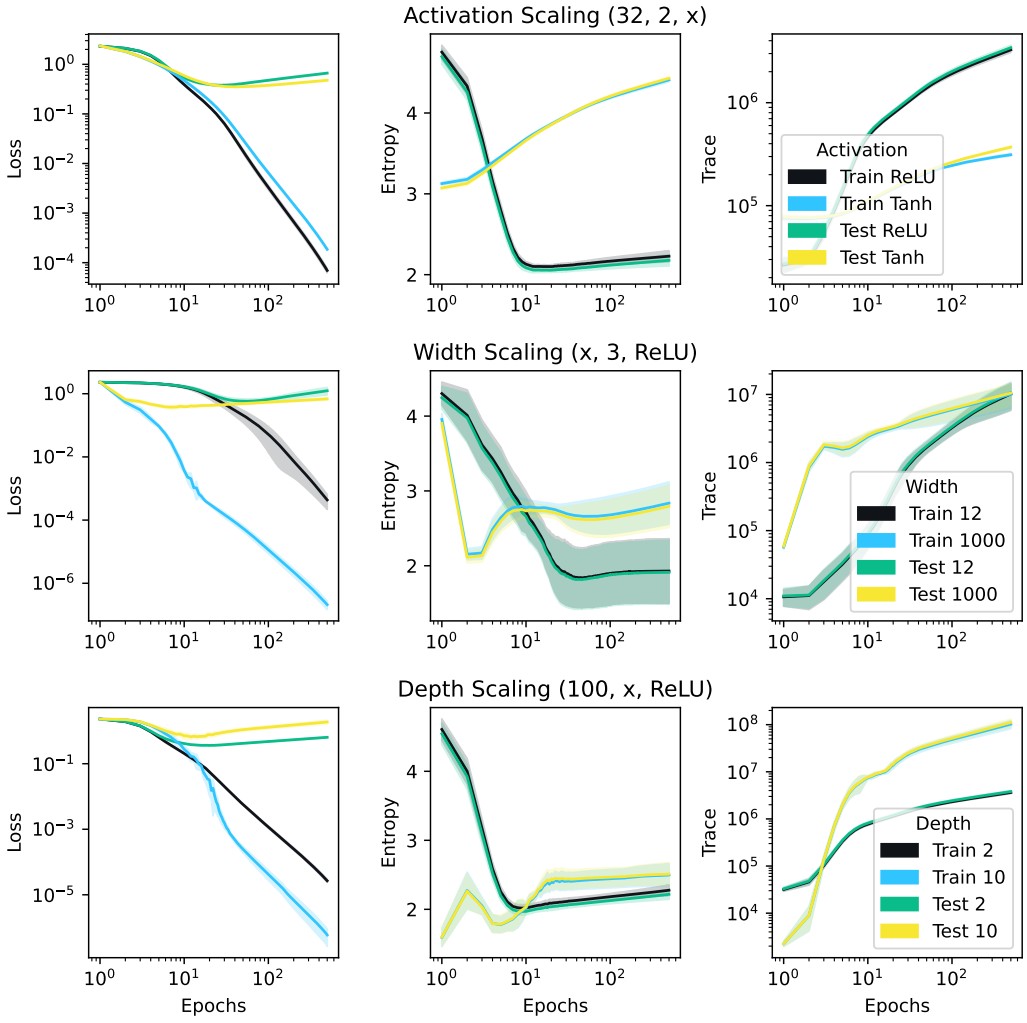

Figure 4: Time evolution diagrams for the dense MNIST study. The top row shows the evolution difference due to a changing activation function. The second row is width scaling, and the third is depth scaling.

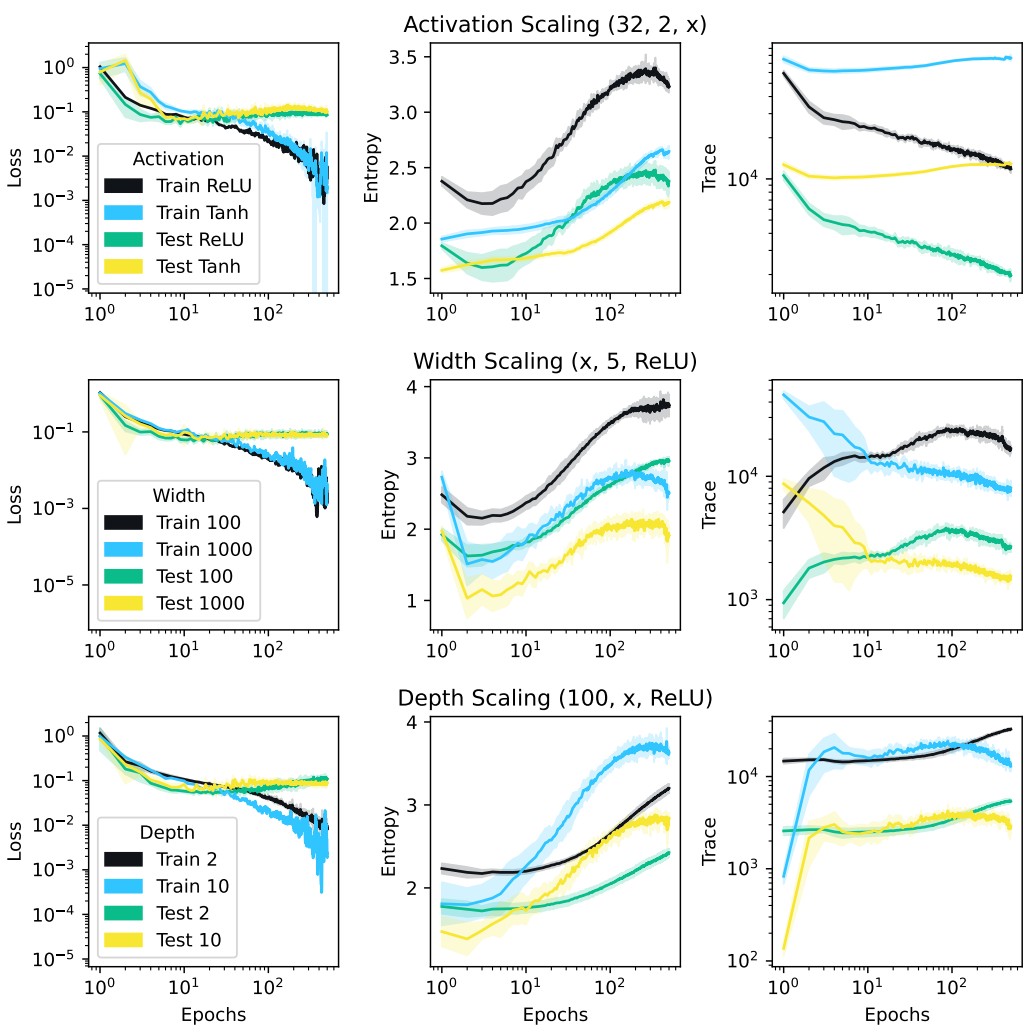

Figure 5: Time evolution diagrams for the dense MPG regression dataset study. The top row shows the evolution difference due to a changing activation function. The second row is width scaling, and the third is depth scaling.

