# OpenReview forum: "Collective variables of neural networks: empirical time evolution and scaling laws"
_ICLR.cc/2025/Conference — Submitted to ICLR 2025_

### Official Review · Reviewer_scGs · 2024-11-01

**Soundness:** 3
**Presentation:** 3
**Contribution:** 3
**Rating:** 8
**Confidence:** 2

**Summary:**

The authors present a study on collective variables of empirical Neural Tangent Kernels (NTK) of various scales, architectures, and tasks in order to describe the learning dynamics of these models. Two major results are described: 1) the evolutions of the entropy of the NTK spectrum during training can undergo two different modes, a decrease in entropy or increase. A decrease in entropy is described as an 'information compression' phase, where gradients from different inputs are aligning. This process seems to occur mostly for smaller networks in both width and depth and is often transitory. Conversely, an increase in entropy is described as ‘structure formation’, where different inputs result in more diverse gradient vectors, indicating more selective representations. An increase in the NTK spectrum’s entropy is almost always observed for large (or over-parametrized) networks and coincides with later stages of training when the models are converging to higher performance. The authors argue that the latter phase of increasing entropy can be argued as when a ‘deep learning’ phase is actually occurring. 2) The trace of the NTK increases as the loss decreases during training, where the trace can be interpreted as an ‘effective step size’ or learning rate. This increase in effective step size, even as the loss decreases, implies a heightened sensitivity of the model to loss, or, as an increase in confidence in the direction of the gradient vectors. The authors argue that this indicate that fictitious data has a stronger negative effect if shown at later stages of training compared to earlier, even if the loss value is the same. Finally, the authors show that these results share some universality across a variety of models and argue that this behaviour might indeed be descriptive of universality in learning dynamics and, in particular, deep learning.

**Strengths:**

Originality: The paper extends known methods, namely computing the trace and entropy of NTK spectra, to a broader scope than previous applications.

Quality: The paper is well motivated, and the background material is introduced with good detail with relevant literatures cited.

Clarity: It is also written clearly, in good scientific English and is relatively comfortable to follow. The key results of the paper are also introduced well and discussed in a way that was easy to appreciate why the authors argue their significance. The figures in the paper are also well formatted.

Significance: The interpretation of the results has the potential to condense the analysis of training dynamics into universal patterns of collective variables, an insight that could help diagnose and analyze a broad variety of models.

**Weaknesses:**

-	(Clarity) For the figures, the colour coding/line-styles aren’t well grouped to my eyes. It might benefit to have the smaller and larger models have similar shades and then have different line styles for train vs. Test. I found myself having to constantly check the legends for each panel which was a bit tedious to interpret each figure.
-	(Significance and Quality) The interpretation of the changes in the entropy of the NTK spectrum as “information compression” vs “structure formation” is, as it’s written, somewhat convincing, but I’m not actually sure if this interpretation is simply a good analogy or if that is what is exactly happening. Perhaps this criticism is due to my own ignorance on the topic, but I think making these statements more convincing would empower this paper quite a bit as these are the central results of the paper. I’m not entirely certain what would make this point more convincing unfortunately, but I wonder if there could be custom datasets that one knows a priori that the data points are somewhat redundant and that the entropy should only decrease versus datasets that require more sophisticated representations and so the entropy should increase, as an example. Doing so might also help relate these results to the results from Tovey et al. (2023) and would help elucidate the contribution. Furthermore, expanding on how these methods are explicitly different from the Tovey et al. (2023) would also clarify the significance and contributions of this paper.

**Questions:**

-	In all the examples shown, it seems the models studied are succeeding (at least somewhat) at solving their tasks. So, we see relatively universal trends, and this is, so far, a reasonable result. However, I wonder what does it look like if the models are too small, or badly initialized, or have the wrong architectures for their tasks? What would these curves look for regimes in which the training dynamics are failing to solve the task?
   - Could you, for example, take some of the models used in figure 3 and handicap them in some way to force them to be bad? What would the curves look like then?
    - Furthermore, for the example shown in figure 3, all these models are large enough such that their entropy is always increasing, as argued by the authors should be the case for over-parametrized, deep models. Could you then make the same study for these models as done in Figure 2 by making them smaller or shallower and recreate the transient decreasing entropy phase?
    - By doing such an analysis, can you relate your results to the cost/performance trade offs for models at different scales?

---

> ### Author Response · Authors · 2024-11-14
> **Response to reviewer**
>
> We thank the reviewer for reading and reviewing the paper in detail and are pleased that they propose accepting it at the conference. We will respond to the comments and questions below. We will first respond to the comments made in the weaknesses report before moving to the questions.
>
> **Comments**
>
> 1. The figures were challenging to put together in a way that made the color clear. We will test alternatives to see if this can be further clarified, as multiple reviewers have commented on this problem.
>
> 2. The use of structure formation comes from the modes of the network aligning into more orthogonal groups, resulting in an increase in the entropy. We can make this clearer in the manuscript. Compression comes from the attempt to make many data points look similar or a focus on learning their similarity. There may be better solutions than these names. One reviewer has suggested structure refinement as an alternative to formation as it aligns well with the traditional view of entropy in physics. We are open to this discussion. The suggested approach provided by the reviewer is also interesting. We had initially considered very simple two-class problems. While they show the same trend, it isn't necessarily more convincing than showing it over the number of tasks and architectures we presented; thus, we went with the current approach. However, a rigorous discussion of what kinds of experiments would further elucidate the point is welcome and expected at such a conference.
>
> **Questions**
>
> 1. The reviewer raises a very interesting proposal. We can show failed models and how their collective variables evolve. As expected, they do not behave according to those above. One issue is that they may show different trends depending on what caused the model failure. While this makes interpretation more challenging, we could explain different mechanisms by which training fails by studying these failures.
>
> 2. We appreciate the reviewer's suggestion to delve deeper into the cost/performance trade-offs associated with large model training. This is a critical aspect that we will address in our future work. We will aim to identify specific mechanisms that drive these trade-offs to obtain simple explanations and guidelines that help practitioners optimize their training process.

---

> > ### Comment · Reviewer_scGs · 2024-11-26
> >
> > Thank you to the authors for replying to the questions/comments.
> >
> > I think this paper is investigating some interesting phenomena and the results are also curious, however I do feel that the interpretation of the data as presented requires a few leaps of imagination (which as others have pointed out, might not be wrong).
> >
> > I won't be changing my main scores, but I will lower my confidence score as I find the comments by the other reviewers are also important/compelling.

---

### Official Review · Reviewer_3ejh · 2024-11-03

**Soundness:** 3
**Presentation:** 3
**Contribution:** 2
**Rating:** 5
**Confidence:** 3

**Summary:**

The authors introduce two quantities to describe the training of neural networks, based on the neural tangent kernel (NTK). These measures are the entropy of eigenvalues in the NTK, and its trace.

They then assert that entropy decreases and increases correspond to "information compression" and "structure formation", respectively. The trace measures aggregated learning rate multipliers over the separate modes of the NTK.

They train multiple neural networks of different types and observe the changes in loss as well as in both entropy and trace of the NTK, trying to identify patterns.

**Strengths:**

Understanding the dynamics of neural network training is an important problem.

The measures are reasonably well described.

The experiments appear thorough.

**Weaknesses:**

It is not quite clear what message we're supposed to get from this work. The authors introduce these measures, and propose some kind of intuitive description of what they measure, which is not necessarily wrong, but what exactly it means for training is not obvious.

The authors also try to discern patterns in the differences in the evolution of these measures with network architecture, but there doesn't seem to be anything consistent beyond "trace increases" and "entropy always increases at some point unless the model is very small". For example,  the authors suggest that smaller networks have only decreasing entropy due to inability to create structure, but then have to admit that this is only for depth, not for width (line 261-262). Even more puzzling are the very large changes caused by changing the activation function, which seem to be at least as large as those caused by architecture changes.

Basically there does not seem to be any clear message from the data.

The parallel with kinetic energy at line 363 is unclear.

The authors make one prediction, if a rather weak one: that introducing noisy data should produce more changes in the parameters at the end of training than at the start of training (line 376). This prediction doesn't seem to be tested.

**Questions:**

Figure 1 is very hard to grasp. Please use similar colors for a given value of the quantity of interest, and separate "test" from "train" with e.g. dotted vs solid, or anything else that would make it easier to separate the values.

Should there be a sum over k in the definition of big-theta_pi in line 133? Where does the k go?

l. 179 is hard to parse.

What are the dark splotches near the sides of the panels in Figure 2?

It's not clear how Eq. 11 was derived. The original equations relate f_dot(xp) to dloss / df, but Eq. 11 show dloss/dtheta_j instead?  What is the j variable here?

---

> ### Author Response · Authors · 2024-11-14
> **Response to reviewer**
>
> We thank the reviewer for reading the paper and for the points they raised. We have tried to address each below. To do so, we address the comments on the weaknesses and the questions in separate sections. As a general comment to the reviewer, it seems that the paper's message was not clear enough. We will use this general feedback to improve its state, although this requires only minor revisions and re-wording.
>
> **Comments**
>
> 1. It may not have come across clearly enough in the manuscript, something we need to address, but we do argue that the entropy increases also onsets with increasing width. This can be seen in the dense neural network studies where this measure is easier to scale. We see smaller networks decrease in entropy, whereas the wider ones increase it.
>
> 2. The role of the trace is best understood by studying its scale effect on the learning rate. This suggests that a trained network is effectively in a different, more sensitive state than an untrained one, something of practical relevance to practitioners. When adding data, for example, in a continuous training paradigm, one may find a disproportionate update to the new data despite its small loss. Carefully selecting batches that offset this would become important. The selection of such a batch may be possible under the introduced framework, and it presents an interesting direction for future study.
>
> 3. The comment regarding the activation function is an open question. As ReLU models are most commonly applied, including in the novelty models presented later, this is a reasonable class to concentrate on. However, the choice of the activation function causing drastic increases in the variables is an open question.
>
> 4. We included the kinetic energy analog as an intuitive explanation of why the trace increases and what this means for sensitivity. But if it is unclear, we feel with minor adjustments to the manuscript that we can concretize this message.
>
> 5. To add to comment two regarding trace, this is not simply the case for noisy data. It could also be overfitting to one mode due to this large trace. We argue that new data points added without a large batch, which would again diversify the gradient, will be trained on more at the end of training than at the start. This scale factor can be very large. So, this does not necessarily apply to noisy data but to any data.
>
> **Questions**
>
> 1. We thank the reviewer for the feedback regarding the initial plots. It is challenging to incorporate all of the information clearly. We will experiment further with colors and line styles to try to simplify this in the final manuscript.
>
> 2. We thank the reviewer for identifying this error. There should be a sum over k in this equation. This will be corrected in the final version.
>
> 3. We thank the reviewer for identifying the difficult to follow sentence in the paper. We feel with minor adjustments this can be further clarified and will make efforts to do so.
>
> 4. The dark patches are failures in the model training, resulting in infinite loss and 0 entropy. We apologize for the lack of clarity. This will be written in the paper and was an oversight in the initial version.
>
> 5. We thank the reviewer for raising this difficulty in the papers. An additional error is that there is a sum over j, which is not clear. One derives the kinetic energy formulation by expanding the derivative via the chain rule and then rearranging the terms.

---

> > ### Comment · Reviewer_3ejh · 2024-11-21
> >
> > I appreciate the author's clarifications. My general appreciation of the paper remains: the paper proposes new descriptive tools and measures which may be of some interest in themselves, but what exactly these measures tell us (in terms of actionable information) remains unclear.

---

> > > ### Author Response · Authors · 2024-11-21
> > > **Response to reviewer**
> > >
> > > In light of the reviewers agreement that these means of analysis provide theoretical and perhaps, practical insight into neural network training, would the reviewer not agree that ICLR is then the right place to present and discuss these ideas

---

### Official Review · Reviewer_vKh1 · 2024-11-04

**Soundness:** 3
**Presentation:** 4
**Contribution:** 2
**Rating:** 8
**Confidence:** 2

**Summary:**

The authors explore neural network learning dynamics and scaling by using recently introduced entropy and trace measures on the empirical neural tangent kernel (NTK) spectrum.
These metrics reveal insights into network representations and their evolution across diverse architectures and datasets, including a toy MNIST problem and more contemporary "novelty" architectures.
Two main mechanisms are seen to emerge (1) information compression, characterized by entropy reduction and (2) structure formation, characterized by entropy increase especially in larger networks (noted by the authors to indicate a "deep learning regime"). While the novelty of the work is somewhat limited (because it's an empirical extension of work by Tovey et al (2023) who introduced the entropy and trace measures), I found the manuscript to be rigorous and very well written.

**Strengths:**

Strengths
* Useful contribution: the problem studied and the point of view presented is practically useful to a large segment of the ML community. The basic claims were well studied/supported, with only minor concerns.
* Paper is clear and well written, and as such has high pedagogical value.

**Weaknesses:**

Weaknesses
* Limited novelty (as mentioned earlier), but by no means a deal breaker
* Some missing details (see below)

**Questions:**

Suggestions for improvement:
* Can the term "deep learning regime" be explicitly defined, and citations provided for wherever this definition matches or deviates from prior literature.
* Please explicitly provide information on network size/architecture/nonlinearities and initialization for every case considered.
* For all tasks, how has the data been preprocessed/scaled/normalized?
* L396: How was "200 samples of 20 data points" chosen? Justify this choice
* L406: Clarify "non-ideal initialization"; what would be ideal?
* L418: Unclear "loss, due to the specific one chosen…". What is being said here?
* L423: Missing details, what algorithm and hyperparameters were used for Deep RL?
* For the RL task, could some classic-control RL tasks be used instead of (or in addition to) a task like Atari CNN
* The idea of kinetic-energy introduced around L367 is quite interesting but seems underdeveloped. It should be left in, but more empirical details about this over training evolution could be provided.
* The authors should consider connecting their findings to previous work on curriculum learning, specifically algorithms where loss on training examples are used for building automatic training schedules.

---

> ### Author Response · Authors · 2024-11-14
> **Response to reviewer**
>
> We thank the reviewer for their reading of the manuscript along with the questions raised and are pleased that they would like to accept it for publication. Below, we answer each question and comment from the reviewer.
>
> **Questions**
>
> 1. The definition of deep learning in the manuscript is a new argument, as the networks have the flexibility to create some structure rather than compress information to solve a task. Upon a literature search when writing the paper, there were few references to the onset of deep learning, with many saying, "More than three layers." However, we can include this reference and some additional citations to papers discussing the onset of regimes in deep learning, describing how they relate to what we have observed in our investigations.
>
> 2. We thank the reviewer for identifying missing information in the manuscript. These details can be added with minor additions to the tables and in small comments in the final manuscript.
>
> 3. We further thank the reviewer for identifying missing training information in the manuscript. The data was normalized for image, generative, and regression tasks, including the GNN molecule model. RL data was used as-is from the game engine, and the language model uses its encoding with no additional normalization. This information can be added to the appendix of the final manuscript in a minor way.
>
> 4. The 200 points are chosen randomly from a uniform distribution of dataset indices without duplication. We found in test cases that this results in a good replication of the full-matrix calculation. Deliberate sampling of the dataset would be an interesting avenue for future investigation.
>
> 5. The reviewer raises an interesting question. By non-ideal, we suggest that the network's initial representations do not capture the structure of the data such that when training begins, these representations first go through some "disentangling" operation where entropy decreases before learning the structure required to solve the problem. Ideal initialization would be one in which inherent structure is better captured or can achieve the required structure more quickly. Performing such an initialization is likely infeasible. Instead, we typically initialize to allow for stable training. However, initialization techniques utilizing entropy and trace are avenues for future investigation.
>
> 6. We apologize for the lack of clarity in the sentence. By specific one, we refer to the specific loss reported from the NLP study, for which several are used to compute the final gradients. The sentence is likely superfluous as we only wish to confirm that the model is trained despite the large loss value. This will be rephrased for clarity.
>
> 7. The reinforcement learning studies used a deep actor-critic model with three-layer convolutional networks. The proximal policy optimization algorithm was used for updates. This information can be included as a minor addition to the appendix of the final manuscript.
>
> 8. The RL component of the study is of great interest to the authors and the idea is to go into more detail in future work about the implications of this result. It would be possible to use more of a standard control algorithm, the cartpole for example, although we are not sure if this would add much to the final message in this particular investigation.
>
> 9. We appreciate the reviewers comment to the kinetic energy argument included in the manuscript. Here, this is to make a physically motivated and intuitive analog to describe trace evolution. Future work focusing more intensely on this physical analog will explore how these principles emerge naturally from machine learning update equations.
>
> 10. We thank the reviewer for the suggestion regarding linking the work to curriculum learning. We will explore the field and add where possible to this and future papers.

---

> > ### Comment · Reviewer_vKh1 · 2024-11-24
> > **Addressed**
> >
> > Thanks to the authors for addressing my feedback. I do not see any need to change my score; but on reading other reviews and the gaps they have found, I have decided to reduce my confidence score to weigh their feedback higher than mine.

---

### Official Review · Reviewer_yWCT · 2024-11-07

**Soundness:** 2
**Presentation:** 3
**Contribution:** 2
**Rating:** 3
**Confidence:** 3

**Summary:**

The authors use entropy and trace of the neural tangent kernel to understand the learning dynamics of neural network esp in regard to architecture scaling. They claim to see initial entropy decrease termed information compression and later entropy increase termed structure formation. They try to identify a deep learning regime. They also show how entropy and trace chance for various architecture-task combinations.

**Strengths:**

The idea of analysing the entropy and scale of the NTK during training seems novel.

The authors have done a fair number of experiments across relevant architecture-task combinations.

**Weaknesses:**

In section 3.1, drawing conclusions on scaling of entropy and trace of the NTK, just from architecture scans on MNIST seems quite premature. When even changing from ReLU to tanh changes results (line 353), it is unclear if results may change based on hyperparams requiring tuning for each architecture, i.e. each architecture needs its own learning rate, initialization, etc. Furthermore, quirks of MNIST might lead to some effects. These conclusions must hold across datasets and architectures to be justified -- the authors show this for dense and convolutional networks and MNIST and Fuel Efficiency tasks and say that results are qualitatively similar. The figures between MNIST and Fuel Efficiency dataset do not seem qualitatively similar to me. Main text says fuel efficiency dataset is Fig. 5 in Appendix, but in Fig 5, it says MPG regression dataset (which is not called this in the table either, but since it is the only task with regressions, this seems to be the Figure - please be consistent in nomenclature) In any case, this doesn't look qualitatively similar to the MNIST figures.

I'm not convinced that the trace gives any extra insight into why malicious data will affect the dataset more in the training. One could just as well use the argument used here in the usual weight update equation. If losses are low, and a new data point has a high loss in particular a high loss whose gradient with respect to current weights is high, then it'll have a large effect later in training.

The lack of a compression phase in models of section 4 is interesting, but is not studied, despite prominently mentioning this as information compression in the MNIST case. The authors just say this may be because of large batch /dataset size. This could have been easily studied by varying the batch size or restricting the dataset in some way.

The trace also varies very differently between different domains and no attempt is made to explain this except some conjectures on missing term in the energy equation.

Minor:
Line 185, use \cite instead of \citep to make authors of a work the subject of a sentence.

**Questions:**

I would like to see very similar behaviour of entropy and trace across a number of tasks to be convinced about this two stage phenomena.

And where it doesn't occur, I expect some effort in trying to figure out why but appropriately titrating aspects of the task / dataset / network between the showing up of two stages vs one stage and figuring out exactly why and when this occurs. Similarly, a better analysis of why the trace shows a dip versus not during learning.

Line 253-254: "increasing entropy indicates structure formation" -- this is counter to the usual physics entropy where entropy decrease with structure formation. I would call the second phase "structure separation" or "structure refinement".

---

> ### Author Response · Authors · 2024-11-14
> **Response to reviewer**
>
> We thank the reviewer for reading the manuscript. We appreciate the reviewer's comments and questions and attempt to answer each one below.
>
> **Comments**
>
> 1. In regard to the qualitative similarity between the architecture scans. The entropy is seen to decrease and subsequently increase in some of the curves in all of the plots shown in the appendix as in the main text. We argue that this satisfies the statement of qualitative similarity.
>
> 2. A point of clarification: In the MNIST architecture scans, the learning rate, batch size, and initialization remain fixed in each training routine. While this is not the best practice when training a state-of-the-art model for deployment, each network learned the task successfully. Therefore, fixing the starting conditions was deemed acceptable.
>
> 3. Apologies for the naming mishap. MPG is an acronym for mile per gallon and is the fuel dataset. This was missed during the review and will be corrected. We thank the reviewer for identifying this error.
>
> 4. We agree to the reviewer's statement that points of high loss cause large gradients. In our work, we want to go beyond this. By computing the trace of the NTK, we isolate the gradient contribution of the network. Even though the gradients of the trained samples are low, the trace becomes very large, which is a feature in the update and acts as an additional scale to the loss.
> For example, with a trace of 1e8, even a loss of 1e-6 will result in a significant update to the network during backpropagation. We point out mathematically that due to model training, future updates will be in a state of heightened sensitivity to new data. This increased sensitivity can be seen as the reason why new points could encounter a large update despite small losses and gradients.
>
> 5. The lack of compression in the larger models is an important point to be addressed. We observed during experiments that these dips were not present without sufficient resolution in recordings. A figure of such an experiment for a network could be included in the final version of the manuscript.
>
> 6. The discrepancy in the trace is a complicated aspect of the paper. A more specific investigation into how this impacts training and why it changes will be performed. We see, for example, that activation functions, in particular, impact the trace. However, while entropy follows a similar trend throughout different tasks, trace does not. This indicates that there is more to trace evolution than simply architecture; thus, a more fine-grained study of the phenomena is required.
>
> 7. We thank the reviewer for identifying the cite command mistake. This has been amended in the new version of the manuscript.
>
> **Questions**
>
> As the first two questions are addressed in the comments above, we will focus on the third.
>
> 1. We chose the term formation because the gradient vectors are necessarily becoming more orthogonal or only aligning along specific directions (related to the modes the network is learning) during training. In this sense, the network is learning the structure of the dataset. However, we agree with the reviewer's statement that this needs to be more intuitive from a standard interpretation of entropy. It is more aligned with entropy in a simple spin-lattice where alignment from increasing energy can nonetheless reduce entropy. In saying that, we are not opposed to changing the term and appreciate the reviewer's suggestion. In particular, structure refinement.

---

> > ### Comment · Reviewer_yWCT · 2024-11-24
> >
> > The authors' response clarifies some of my concerns, in particular their point 4 clarifies what they mean about the heightened sensitivity to new data after training due to large trace. However, despite these clarifications, most of their claims are not shown convincingly enough via experiments, and to my mind remain conjectures.  Overall, their ideas seem promising, but still need targeted experiments to demonstrate.

---

### Meta-Review · Area_Chair_BoHK · 2024-12-08

**Metareview:**

This submission investigates properties of the empirical neural tangent kernel (NTK), and demonstrates that the properties change predictably over the course of learning in two qualitative phases.

Empirical studies of the NTK may provide insight into the learning dynamics of models that follow NTK dynamics. However, I concur with Reviewer 3ejh that the descriptions afforded by the NTK trace and entropy are correlational in that they do not provide more than quantities that correlate with loss, even if they do so predictably, and that the phases identified are also observational rather than causal properties of the dynamics. To make the case as claimed in the manuscript that this empirical analysis "enhances our theoretical understanding of neural network dynamics and provides practical insights for optimizing architecture design and training protocols," the authors should answer the question of what to do with these descriptions—for example, to answer outstanding questions in neural network theory or, more practically, provide a way to attenuate a certain downstream property—which is not yet done in the manuscript.

Additionally, the authors nowhere discuss deviations of neural network training from the NTK regime; see [Geiger et al. (2019)](https://arxiv.org/abs/1906.08034) for example, and the discussion on μ-transfer confounds these regimes. Since feature learning is attested to underlie deep learning (see [this blog](https://kempnerinstitute.harvard.edu/research/deeper-learning/infinite-limits-of-neural-networks/) for a pedagogical overview), this may limit the explanatory power of any quantities derived from the NTK, including those derived here (even if reevaluated empirically throughout training).

**Additional Comments On Reviewer Discussion:**

#### Reviewer yWCT
Asked for concrete predictions on why or why not the quantities introduced follow the observed trajectories, which remains unanswered. Questioned whether results were truly consistent across architectures and datasets. While authors clarified some points about model sensitivity, the reviewer maintained that many claims remained conjectural and needed more targeted experiments.

#### Reviewer vKh1
Found the work rigorous and well-written despite limited novelty. Requested primarily clarification on technical details and definitions. Reduced confidence after seeing other reviewers' critiques but maintained positive rating.

#### Reviewer 3ejh
Emphasized that while the measures were interesting, their practical significance remained unclear. Questioned the physical analogies and mathematical derivations. After author responses, maintained that the paper provided descriptive tools but lacked actionable insights.

#### Reviewer scGs
Appreciated the potential universal patterns in training dynamics but wanted more exploration of failure cases and scaling relationships. While supportive overall, reduced confidence after seeing other reviewers' concerns about interpretability and practical utility.

#### Finally
The authors responded to specific technical questions and offered to provide additional details and clarifications. However, the core concerns about the practical utility of the findings and their theoretical significance remained unresolved. The Area Chair ultimately sided with the more critical assessments, emphasizing that the paper's correlational findings, while interesting, do not yet sufficiently advance theoretical understanding or provide practical insights for improving neural network training.

---

### Decision · Program_Chairs · 2025-01-22

Reject